artificial intelligence

tornado, debris, structural damage, machine learning,

**Author for correspondence:**
Hussam N. Mahmoud
e-mail: hussam.mahmoud@colostate.edu

# Interpreting the socio-technical interactions within a wind damage–artificial neural network model for community resilience

Stephanie F. Pilkington[1] and Hussam N. Mahmoud[2]

[1]University of North Carolina at Charlotte, Charlotte, NC, USA
[2]Colorado State University, Fort Collins, CO 80523-1372, USA

HNM, 0000-0002-3106-6067

The use of machine learning has grown in popularity in various disciplines. Despite the popularity, the apparent 'black box' nature of such tools continues to be an area of concern. In this article, we attempt to unravel the complexity of this black box by exploring the use of artificial neural networks (ANNs), coupled with graph theory, to model and interpret the spatial distribution of building damage from extreme wind events at a community level. Structural wind damage is a topic that is mostly well understood for how wind pressure translates to extreme loading on a structure, how debris can affect that loading and how specific social characteristics contribute to the overall population vulnerability. While these themes are widely accepted, they have proven difficult to model in a cohesive manner, which has led primarily to physical damage models considering wind loading only as it relates to structural capacity. We take advantage of this modelling difficulty to reflect on two different ANN models for predicting the spatial distribution of structural damage due to wind loading. Through graph theory analysis, we study the internal patterns of the apparent black box of artificial intelligence of the models and show that social parameters are key to predict structural damage.

## 1. Introduction

The creation of predictive models using machine learning has grown in popularity in recent years due to the ability to model complex, nonlinear, relationships. Machine learning approaches have proven viable within the field of natural hazards. Previous

studies have applied such techniques to enhance modelling, forecasts and potentially predict outcomes or impacts from hazards [1–8]. Each of these studies used artificial neural networks (ANNs), a type of machine learning (ML), in which the ANN is best used for modelling a singular problem, for example, forecasting the overall impact, in terms of economic damage, from a hurricane event [1,9,10]. Similarly, extreme wind hazards have the potential to cause damage to, or even completely destroy, a structure. The current modelling approach of such damage is based on wind speed and the probability of failure of structural components under wind pressure. The overall knowledge of the physical characteristics of a structure and how it reacts to wind loading has been well established and incorporated within building codes [11]. However, there are factors outside of structural engineering that, while accepted as potentially contributing to damage, have proven difficult to model alongside the structural characteristics. Such factors include debris impacts and social characteristics. One of the aforementioned studies [5] specifically evaluates the applicability in creating an ANN to predict property damage, in terms of economic loss, from tornadic wind events. The ANN model was built using data related to tornado intensity, land cover and a significant number of potential sociological vulnerability demographics. While the authors demonstrate viability for ANNs to be applied in determining tornadic damage and loss, they do suggest additional research to further refine the approach. Additionally, their model does not consider structural variables, whereas the field's current physics-based modelling approach to wind-induced damage (and subsequent loss) is solely structural engineering based. The ANN model discussed herein uses relevant tornado, societal demographic and structural data to determine a building's resulting damage state from an extreme wind event.

Even though popularity in applying ANNs has risen, concern remains over the apparent 'black box' type structure of artificial intelligence (AI). From a modelling standpoint, it would appear that data is 'fed' into the model without much understanding of how the resulting outputs are reached even if modelling errors are low overall. This raises further concern regarding whether or not correct patterns are being established and how each input variable is related within an entire network. Since these modelling networks are designed with the intent to mimic neural connections within the human brain, it may be possible to interpret ANNs as we would in uncovering patterns within a human brain. Within the research presented herein, ANN models designed to predict a resulting building damage state from extreme wind event will be unravelled. Specifically, two ANN models will be primarily referenced and explored herein: (i) one with inputs related solely to the wind hazard and the engineered structure and (ii) another with inputs related to the hazard, structure, its use (social characteristics in the form of tenure and total population) and potential for debris within the surrounding area (in the form of per cent area forested and the housing density). Interpreting both of these ANNs using common mathematical treatment of graph theory (e.g. shortest path) may allow for potential insight into what connections are occurring to relate the various input parameters (e.g. wind hazard, structural parameters and social characteristics) to outputs (structural damage) within the ANN 'black box' and how social and engineering characteristics are affecting an overall resulting outcome.

In this paper, we first, in §2, provide a high-level overview of ANNs and highlight one of the major issues with their use in different disciplines, which is the ability to physically interpret the relationship between inputs and outputs in well-trained networks. In §3, we provide background on graph theory, its overall use in different engineering and medical fields, and its possible use for unravelling the connections between input variables and outcomes within an ANN. In §4, we focus on the research conducted with an emphasis on data assimilation, building the ANNs and selecting the ANN models for analysis. We finally provide the discussion of the results in §5 followed by general conclusions in §6.

## 2. Overview of artificial neural networks

ANNs fall within the purview of artificial intelligence by attempting to mimic how the human brain learns through data analysis and pattern recognition. In essence, this is accomplished through various forms of regression analysis given a historical dataset. This historical dataset consists of multiple input variables that could be tied to a specific desired output (or outcome). A 'supervised' network, as was evaluated herein, includes data with known outputs and inputs used to 'build' (train, validate and test) the network. A typical feed-forward ANN, as used herein, consists of connections between neurons that can either be activated or deactivated and can vary in strength through calculated connection weight values. Within the brain, it is not necessary that every single neuron connects to every other neuron, but connections within will establish paths through taking input stimulus information to other neurons that eventually

lead to exciting action within a person. How this is accomplished mathematically for ANNs, through various training algorithms in backpropagation, is outlined in detail within the Methods section.

In building an ANN, training establishes the connection patterns (weights) between neurons, which are then validated for lowest possible mean square error (m.s.e.) and subsequently tested. The ANN type (training theory used, number of hidden neurons and layers, and so on) is dependent on the type of problem the designer is trying to solve and leads to a versatile application of ANNs while still appearing to be an accurate model. Despite the apparent modelling success on the surface, there is a concern as to understanding how these networks reach such conclusions [12]. Typically, in understanding an ANN, a secondary model, of a different nature (not AI or ML), is used to explain the model results. This is termed 'Explainable Machine Learning'. There is also a concept of 'Interpretable Machine Learning', which proposes the addition of application-specific constraints on the ANN model [12]. However, this approach is typically forgone as the problems usually addressed with ANNs are especially difficult to solve conceptually [12]. Ultimately, what such concerns highlight is a need for model transparency through interpretable ML. This becomes specifically concerning when considering that multiple different ML algorithms could produce similarly acceptable results [12], leading to the question of 'how do we truly *know* which model is producing the most conceptually accurate results?' *While there is a large trend in evaluating graph theory models using ANNs, we propose that a way to answer this question is by using graph theory as a means of analysing an ANN's patterns and neural connections.*

# 3. Analysing artificial neural networks with graph theory

Graph theory is a commonly used tool in modelling and understanding complex networks. For example, travel time options between a location and a destination are determined using graph theory methods by evaluating all possible routes or paths. The brain has also been well known to constitute a complicated structural network consisting of nodes, or neurons/cells, that connect through synapses [13,14]. Neurons within a certain portion of the brain are also thought to connect to other portions of the brain based on spatial proximity [15]. The topological structure and synapse lengths have been evaluated as potential factors in many cerebral type diseases. Previous studies related to the human brain and conditions such as Alzheimer's [16–18] and Schizophrenia [19–21] have used graph theory to analyse magnetic resonance images (MRIs), or functional magnetic resonance images (fMRIs), and electroencephalograms (EEGs) of patients with such conditions and patients without.

Each node within the created association matrix graph may be evaluated for its degree, which is the number of connections either entering or exiting a node [15,22]. A high in-degree would show that a specific node is affected by many other nodes, or attributes. A high out-degree value would show that a specific node affects many other nodes, or attributes. In other words, it may contribute to the outcome of many subsequent nodes [22]. The paths between the two nodes can indicate the strength of association between two nodes (path length) or the level of redundancy between two nodes (number of paths) [22]. Connection density within a graph is considered an estimator of physical cost in that it is a ratio of the number of connections present to the number of possible connections between nodes [15]. Another commonly evaluated graphical analysis is the determination of hubs, which are nodes with high centrality, indicating that a specific node is vital to communicating data across the network and how shifts in centrality within the network structure may lead to different outcomes [15]. For example, Alzheimer's patients [19,20] showed lower centrality within the hippocampus than those without.

The studies provided a starting point for assessing ANN models through similar graphical analysis. Instead of MRIs and EEGs, data were collected on one specific modelling problem (outlined further within the Methods section), and the association matrix was subsequently related to the ANN connection weights that are established within the build processes. If the studies above highlight sections of the brain that perform specific tasks, then an ANN would be similar to a cluster of neurons within that area. In other words, as shown in figure 1, the same graphical analysis approach used to assess how sections of the brain interact for different patients could potentially be used to evaluate interactions between individual neurons.

Path length within an ANN would now relate to an accumulation of connection weights as data moves from an input, to hidden, to output neurons, with the interest focused on which paths through the network produced large values, indicating higher weights and stronger connections from a specific input to a specific output. Clustering, closeness, or centrality, within a network will now

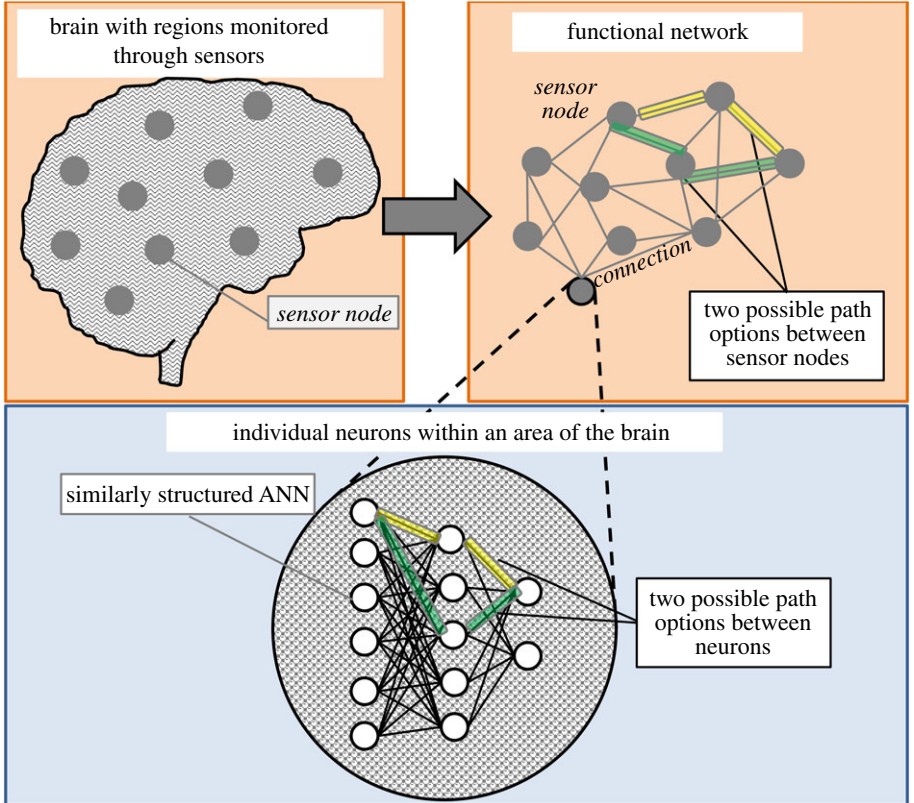

**Figure 1.** Conceptual depiction of how brain data, monitored through sensors, translates to a complex graphical network and how individual neurons fit into this structure.

assess how well connected a single input or output neuron is to the rest of the network, where a high closeness (or centrality) indicates a network that is either highly influential to the network (input) or is widely influenced by many parameters (output). The definitions of such concepts, in relation to how they could be applied to a single hidden layer feed-forward ANN, are provided in the Methods section. These graph theory concepts are similar to those used to evaluate patients with, or without, Alzheimer's or Schizophrenia. This leads to an interesting possibility in interpreting ANNs using similar methods for understanding the brain, especially since ANNs were originally structured based on brain functionality.

## 3.1. Shortest path

The shortest path problem considers the minimum 'length' to travel from neuron $j$ to neuron $i$. A common example of this concept would be to consider two buildings with multiple road routes from building A to building B. Each road has a travel time associated with it, which would be equivalent to a weight value. The shortest path problem assesses the shortest travel time between A and B [23]. With an ANN, the travel time was considered equivalent to the weights, such that the shortest path ($P$) is defined as

$$P = \text{Min}[w_{ij}^{\text{input}-\text{hidden}} + w_{ij}^{\text{hidden}-\text{output}}], \tag{3.1}$$

where the possible path summations were simply the possible routes information can flow from input to hidden and then hidden to output layers and the corresponding weights ($w_{ij}$) between neurons. Higher value weights, in this case, were considered stronger connections between neurons; therefore, the results of this analysis allowed for an evaluation of how strongly an input correlated to the desired output.

## 3.2. Centrality: degree

The degree of a neuron is considered as the number of connections associated with said neuron [24]. This was further subdivided into how many of those connections were 'entering' the neuron, or the 'in-degree'

$(D_{in})$, and how many were exiting the neuron, or the 'out-degree' $(D_{out})$, such that

$$D_{in} = \sum e_{ij}^{(p \text{ or } n)} \tag{3.2}$$

and

$$D_{out} = \sum e_{ij}^{(m \text{ or } p)}, \tag{3.3}$$

where in-degree could only be calculated for a connection $(e)$ between a hidden $(p)$ or output $(n)$ neuron, and out-degree could only be calculated for an input $(m)$ or hidden neuron $(p)$ in a feed-forward network. By using graphical analysis, a single neuron could be isolated for its contribution to the network (inputs) or its reliance on data translated through the network (outputs), such that higher out-degree values indicate that an input neuron heavily contributes to the overall network and higher in-degree values indicate that an output neuron is heavily influenced by many neurons.

## 3.3. Centrality closeness

Closeness is another way to measure a neuron's centrality within the network. This form of centrality measures the inverse sum of the weights from one neuron to all other neurons such that

$$C = \left(\frac{A_x}{Q-1}\right)^2 \frac{1}{\sum w_{ij}^x}, \tag{3.4}$$

where $A_x$ is the total number of reachable neurons, divided by $Q-1$, where $Q$ is the total number of neurons within the network, squared then divided by the sum of all reachable connection lengths (weights) from a specified neuron $x$. For in-closeness, the focus of this analysis was on the output neurons, as these are the factors with data feeding 'in' to them. Conversely for out-closeness, the focus was on the input neurons. Closeness values were meant as another means of measuring network contribution similar to degree.

# 4. Methods

## 4.1. Assimilation of data

To create the models discussed in this article, data assimilation was necessary from two very different subject areas (engineering and social vulnerability) at the time of a wind event. Indeed, the validity of the approach is reliant on what data is available. The National Weather Service (NWS) maintains publicly available records from damage surveys related to wind events [25]. These records typically comprised building damage description, location, wind speed, date of event and, often, a photo of the damaged structure. From this, the building construction (materials and geometries), hazard parameters and damage state of the building were gathered and organized in a text file. The US Census Bureau tracks many population characteristics, including factors related to social vulnerability [26–28]. These data can be spatially organized by (in decreasing area size) county, census tract, block group and block.

In this study, 117 data points were collected. Each building data point was associated with structural and social characteristics as illustrated by example, for selected features, in figure 2. Building structural characteristics include median year built for buildings in the specified block group (from US Census), occupancy class, main wind force resisting system (MWFRS) material and the exterior façade materials (both wall and roofing), the roof shape, number of (equivalent) storeys and the footprint area. These data variables were all chosen based on generally accepted factors used in calculating wind resistance of a structure [11,30–32] and were mostly gathered from building images geotagged by an NWS damage survey. These buildings then assume the census block group demographic characteristics for its location. Of all available census data, the demographics were chosen based on the vulnerability factors outlined in various sociological studies [26–28,33–36]. For this study, a preliminary dataset was created for the state of Missouri as a baseline for determining what factors intertwine best to determine building damage. Missouri was largely chosen for its location within Tornado Alley and so that the models built could be validated against known data from the 2011 Joplin tornado. The full compiled Missouri data, the key for how these items were coded, and an overview of the comparable performances of multiple model variations are available in Section S1 of the electronic supplementary information.

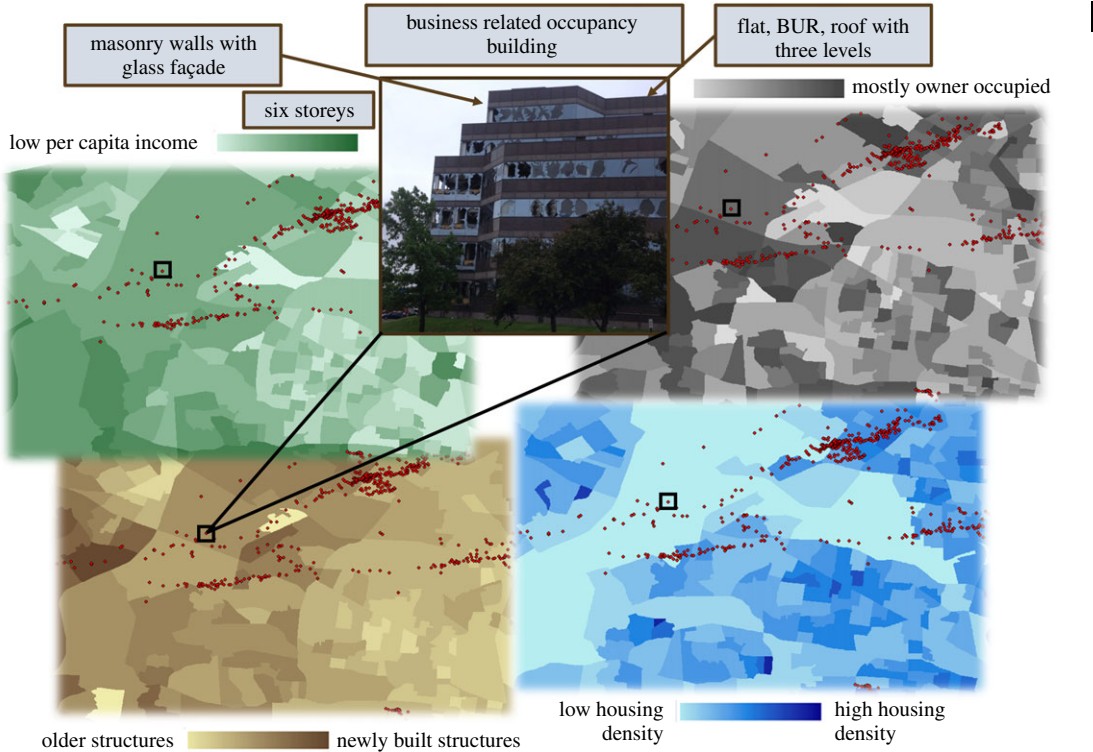

**Figure 2.** Building data points (red) spatially correlated to selected census block group data [29] shown by example of one building in the St Louis metro area and selected societal factors for 101 mph winds. (Building photo is from NWS damage survey [25]).

## 4.2. Building artificial neural networks

Once the data were gathered, the ANNs could be built and validated. Determining the structural layout was the first step in building an ANN, and within this research, a 10-hidden neuron structure was used as it fell between the number of outputs (5 neurons) and number of inputs, which varied across the multiple models being evaluated. There were 117 datasets used to train the ANN models as noted in §4.1. Of those data points, 70% were used in the training, 15% in the validation and 15% in the testing phases of building each ANN for most training algorithms. Model (A) and Model (B), however, used Bayesian methods for building, in which no 'validation' phase was used, leading to using 90% of the data in the training phase and 10% in the testing phase. However, both of these models went through a hindcast validation, in which each would model building damage for the 2011 Joplin tornado and compare the results to actual collected data from storm surveys following the event and is discussed further on within this section. Note that data from the Joplin tornado was not included in the dataset used to build these models.

Each connection within the model networks had an adjustable weight value ($W$) that would dictate if a connection was activated, and if so, how 'strong' of a connection. The hidden and output neurons also had bias values ($b$) acting on the neurons themselves. The dataset ($D$) used to build these model ANNs consisted of inputs (related to the wind hazard, structural characteristics and socioeconomic data) and outputs (resulting damage state). However, within this dataset, the scales for each input variable differ. For example, wind speed ranged from 0 to 175 knots, while the median year built ranged from 1800 to 2018. The use of a transfer function (s-curve) served as a mean to normalize data before entering the next layer, such that each input had the same bounds. In keeping with this concept, there was a transfer function associated with the inputs before entering the hidden layer, and the hidden neurons before entering the output layer. The most common forms of transfer functions are the log-sigmoid, tan-sigmoid and a pure linear function. The pure linear function was not used, as the desirability of using an ANN is its nonlinear nature. This research, along with most other classification type problems, used target values that are binary. This may seem to suggest that a log-sigmoid transfer function would be ideal as it bounds the asymptotes to zero and one. However, choosing this function would cause weights to become 'stuck'. The training process would attempt to

fit the output data as close as possible to the target values (causing over-fitting), which would only be achieved asymptotically, causing large weight values that would eventually result in a gradient that produces weight update values to be close to zero [37]. Therefore, the tan-sigmoid function was used as the transfer function in building these ANNs. Both the log-sigmoid and tan-sigmoid functions are bound by [0,1] and [−1,1], respectively, as outlined by the following equations:

$$f(s_i) = \frac{1}{(1 + e^{-s_i})} \tag{4.1}$$

and

$$f(s_i) = \frac{2}{(1 - e^{-2s_i})} - 1. \tag{4.2}$$

where $s_i$ is each neurons' activation (or voltage to excite), ranging from infinity to negative infinity, such that

$$s_i = b_i + \sum_{j=1}^{i-1} W_{ij}x_j, \tag{4.3}$$

where $W$ represents the weight from the $j$th to the $i$th neuron, $b$ is the bias on the $i$th neuron, and $x$ relates to the neuron values being 'fed' into the following layer [38,39]. Within each network build, the weights and biases were adjusted during each iteration. At least six iterations were conducted, and if the first of the six proved to produce the lowest possible error, then that was the resulting end network that moved on to the testing phase and was subsequently applied. If not, then the weights and bias adjustments continued until the lowest possible error was reached between the output and target values using standard backpropagation in a multi-layer network. ANNs could be trained by minimizing the mean square error (m.s.e.) or sum square error (s.s.e.), $E$, between the network output value, $y$, and the actual target value (known from dataset), $Y$, as computed by

$$E(w_{ij}) = \frac{1}{z} \sum_{i=1}^{n} [y_i - Y_i]^2, \tag{4.4}$$

for $n$ number of output neurons (for example, damage states 0 through 4) with $z = n$ for m.s.e. and $z = 2$ for s.s.e. approaches. The ANNs analysed herein assessed the error using m.s.e., and comparisons of the two approaches can be reviewed in Section S2 of the electronic supplementary information.

Subsequently, the target value, $Y$, was a function of the dataset $D(x,y)$ and the activation function, $s$. The error determined in equation (4.4) was propagated back through the network to adjust the weights and biases as well as change the neuron activations by means of a myriad of training algorithm approaches. An analysis of approximately 10 algorithms [40–49] was conducted and Bayesian regulation (BR) was determined to produce the lowest per cent error during the build process, the details of which can be found in Section S2 in the electronic supplementary material, Information. This training algorithm is probabilistic focused in Bayes theory, instead of using gradient descent type of approaches. In other words, BR evaluates the probability of the weights, $w_{ij}$, on the connections between neurons given the dataset $D$ [48,49] such that

$$p(w_{ij}|D_i) = \frac{P(w_{ij})P(D_i|w_{ij})}{P(w_{ij})P(D_i|w_{ij})}. \tag{4.5}$$

## 4.3. Choosing artificial neural network models for evaluation

Models (A) and (B) differ only in the input variable data and therefore the number of input neurons. Both models use Bayesian training to adjust the weights and biases for an ANN of one hidden layer consisting of 10 hidden neurons. Models (A) and (B) are two of a total of 10 possible models each created with different inputs considered. Included in these 10 models was one model that also went through 10 training algorithm variations, each for m.s.e. and s.s.e. measurements, with the goal of choosing the best possible training algorithm for modelling damage state. The determination of Bayesian over other possible algorithms and the ultimate use of Model (A) as the 'best performing' model was conducted through a series of ANN builds to reach a lowest m.s.e. or s.s.e.. This was done to determine the statistical distribution of possible errors given each algorithm or each model. Ultimately, the per cent error (number of parameters misclassified through all phases), training performance (how well the

network was able to minimize error through backpropagation), false positive rate (FPR), false negative rate (FNR), true positive rate (TPR) and true negative rate (TNR) were tracked. The FPR, FNR, TPR and TNR are associated with the receiver operating characteristics (ROC) of the network, which was why they were deemed valuable parameters to judge the networks. The per cent error is essentially an overall result of these rates. As an example, the FPR is known as the number of false positives associated with the produced network. For example, a false positive is similar to the 'crying wolf' outcome where it is said a 'worse' result will occur, but it does not and can be given by the equation

$$FPR = \frac{\text{false positives}}{\text{all output positives}}. \tag{4.6}$$

The remaining rates are given by the following equations:

$$FNR = \frac{\text{false negatives}}{\text{all output negatives}}. \tag{4.7}$$

$$TPR = \frac{\text{true positives}}{\text{all output positives}} \tag{4.8}$$

$$TNR = \frac{\text{true negatives}}{\text{all output negatives}}. \tag{4.9}$$

The 'true' rates were desired to be closer to one (or 100%), while the 'false' rates were desired to be closer to zero. The initial model was built a minimum of 50 times for each training algorithm outlined above. The data from these builds were then assimilated in order to determine the mean, mode, maximum, minimum and standard deviation for each performance characterizing indicator (PCI). FPR, FNR, TPR and TNR were originally calculated for each 'class', for example, the damage state classification neuron, but the mean of all the five classes was used in the evaluation of the PCIs in the interest of assessing the accuracy across the whole network. In order to ensure this was an adequate representation of each algorithm, the PCI data from multiple builds was evaluated for a coefficient of variation of less than 0.5 with 95% of the data falling within ±2 s.d. (two standard deviations). In other words, the majority of builds did not vary by more than 50% of the mean, making the sample size representative of the most likely PCI outcomes. If 50 builds did not result in meeting these requirements, then additional builds were created until the PCI data fell within these requirements, resulting in builds for certain algorithms counting in the hundreds. The 'best performing' Model (A) builds were determined to be those with lower per cent errors (typically associated with a lower mean square error), FPR and FNR, but higher TPR and TNRs. From this point, those best performing ANN methods (training algorithms and performance functions) were used in the subsequent step of altering the input parameters.

The results of these statistical distributions are shown in detail in Section S3 in the electronic supplementary material, information. Ultimately, this analysis of the range of performance for different training algorithms and subsequently different model inputs resulted in an initial choosing of Model (A) as a better performing model and Model (B) as one of the poorer performing models. However, in the interest of further confirmation, both Models (A) and (B) were simulated for the 2011 Joplin tornado and compared to the known results. Figure 3 shows a comparative analysis at the building level for an exact match to the known resulting damage state for a specific building (as tallied from post-event video data [50]). An additional comparison was evaluated for an overall approximated match, in which a total count of buildings within lower damage states (DS1 + 2), as provided in post-event surveys [51,52], was compared to the amount categorized in either of those damage state for the model simulations. The same was also done for the damage states that would be considered a total loss for insurance purposes (DS3 + 4). Ultimately, Model (A) and Model (B) showed approximately 41% accuracy and 44% accuracy, respectively, when individual building damage states were compared. However, when compared to whether or not buildings community wide were a total insurance loss, Model (B) had an 87% accuracy for those in DS1 + 2 and a 90% accuracy for those in DS3 + 4. Whereas, Model (A) had approximately 60% accuracy for DS1 + 2 and 70% accuracy for DS3 + 4. This additional hindcast validation served as a means to support that Model (B) could be considered the 'better performing' ANN Model. Note that with the hindcasting procedure, the ensemble of multiple ANNs was used to constitute each model.

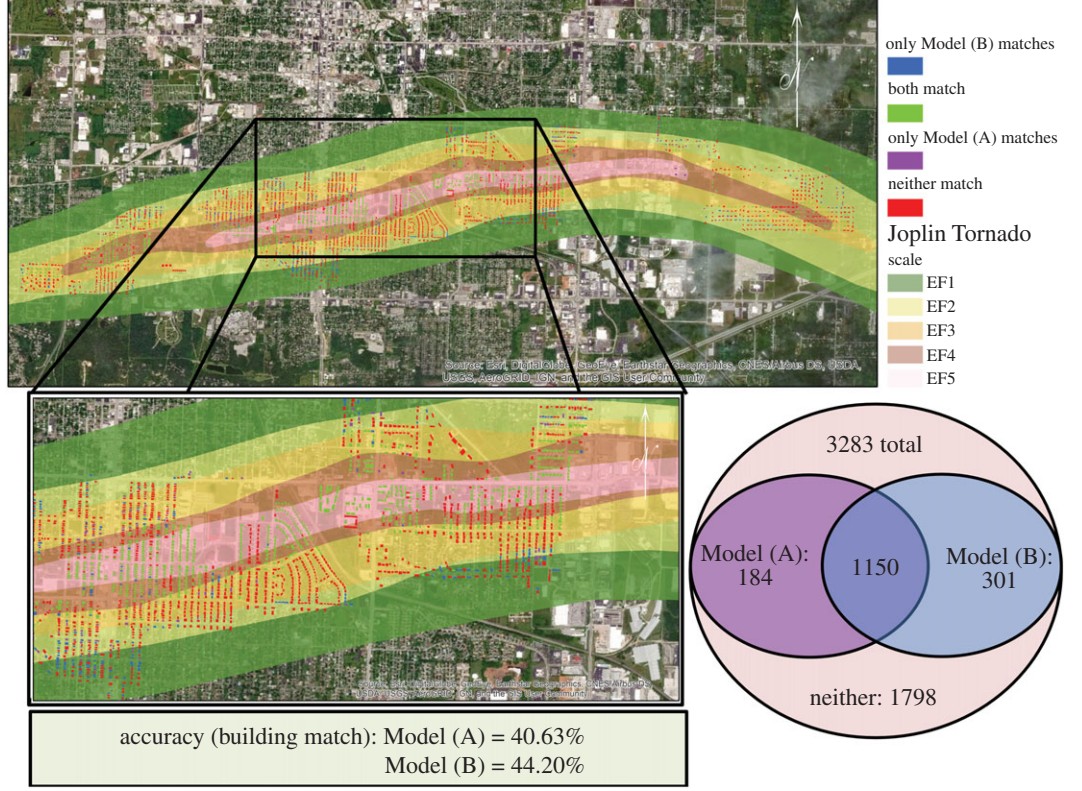

**Figure 3.** Results in matching an individual building's damage state to that determined using Model (A) and Model (B).

## 4.4. Final wind damage–artificial neural network models

Ten ANN models were developed as listed in the electronic supplementary material, table S1 and table S1-A and discussed in detail in Section S1 in the electronic supplementary material, two of which are discussed herein—Model (A) and Model (B). Model (A) is named Damage Model 3 in the electronic supplementary material and pertains only to the structural code characteristics of ASCE 7–10 (and ASCE 7–16). Model (B) is named Damage Model 8 in the electronic supplementary material and in addition to the structural characteristics, it includes census block group population count, housing tenure (per cent own and per cent rent), housing density, population density, per cent of surrounding area forested and per cent of the surrounding area with impervious surfaces as noted in the electronic supplementary material, table S1. The outputs to these models are the building damage state, which ranges from DS0 to DS4 [30,53], as illustrated in figure 4. The data used to build such ANNs consists of historical events with post-storm survey data from the US National Weather Service (NWS), including images [25,54], and social data from US Census [55] as further detailed within the Methods section.

In building these ANN models, Model (A) was shown to have performed slightly poorer than Model (B) during the build process (network mean square error (m.s.e.) and per cent error). When both models were used to hindcast community-wide building damage resulting from the May 2011 Joplin tornado, Model (B) performed observably better, leading to the suggestion that incorporation of the social characteristics and debris potential variables listed above was a more accurate modelling approach. A subsequent sensitivity analysis showed that when all other variables remained constant, increasing debris potential increased the resulting building damage state, while a decrease in both renting and owning tenure (increase in 'other' tenure) showed an increase in damage state.

The results of this analysis were to be expected based on knowledge obtained from various other fields related to wind hazards and community vulnerability and resilience. Many studies have evaluated how debris can puncture a building [56,57], and it is widely accepted that once a hole is created, the wind load-based pressure surrounding the structure changes. When this occurs, for example, the pressure 'pulling' on the roof from the exterior combines with an additional pressure force from wind that has entered the structure, which is now 'pushing' upward on the roof from the

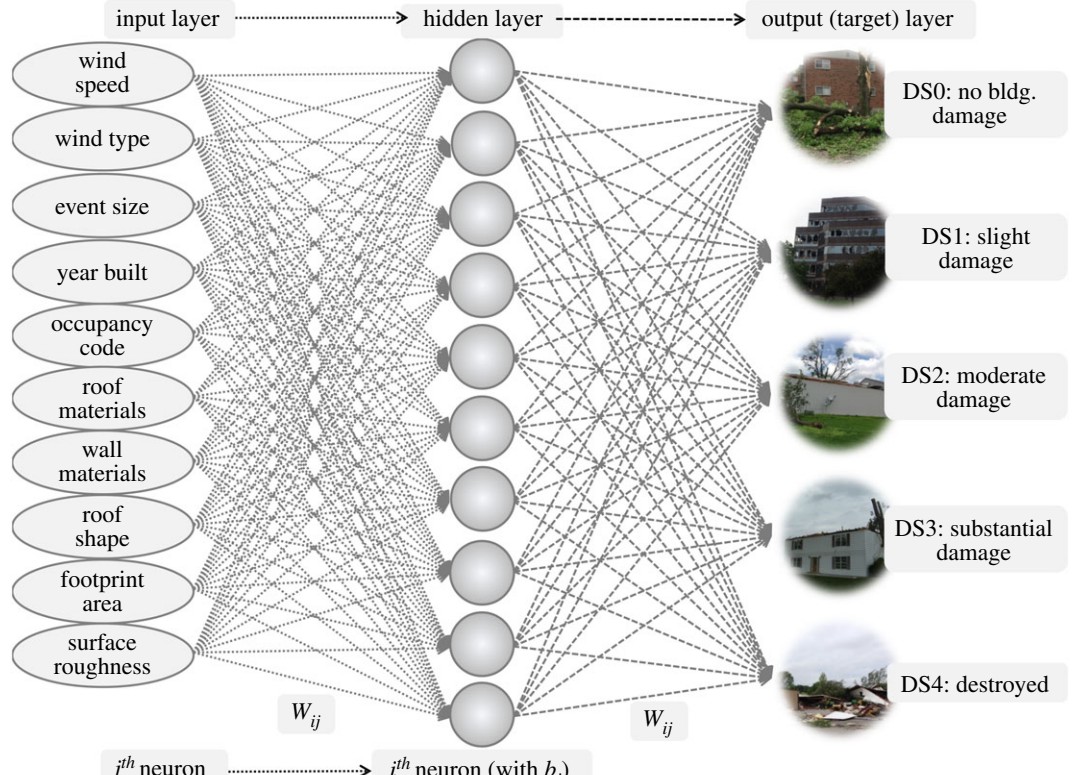

**Figure 4.** Model (A) ANN structure with input and output variables (building damage images from NWS Damage Survey Viewer [25]).

interior of the building. However, in modelling on a grand scale (entire wind event area affecting many community buildings), such small-scale fluid dynamic interactions have proven difficult to capture. Additionally, many social vulnerability studies [26–28] highlight tenure, as well as other demographics, as factors that may either increase or decrease the vulnerability of a population in severe events. However, vulnerability may not precisely translate to structural damage, which creates an additional challenge in integrating social and engineering-based factors when modelling community impact, recovery and resilience from natural hazards.

Essentially, these ANNs may provide a means to model variables across multiple disciplines that have proven challenging to coalesce in a physics-based modelling approach. This presents a unique opportunity in evaluating how these ANNs integrate such variables in determining a resulting damage state, since it is well known that these additional variables contribute to damage in some way. Since, as was previously stated, many ANNs can produce similar error values during their builds, multiple ANNs were used within this evaluation, which resulted in errors ranging from 3.4 to 6% error for Model (A) and 1.7 to 4.3% error for Model (B), as these were in the lowest error ranges possible for each model using Bayesian training methods, totalling six ANNs for each model. In evaluating the network structures using graph theory, two approaches were taken: an averaged approach, in which each ANN was evaluated individually then averaged across that model type, and a combined approach, in which the model ANNs were all linked together through the incorporation of each ANN's hidden layer connections.

## 5. Discussion of results

The shortest path analysis allowed for observations of variables that may be strongly connected to specific damage states. For this analysis within an ANN, the interest focused on higher connection weights. Therefore, if the shortest possible path from one neuron to another was a relatively high value, then any other path options within the network would be equally, or more, strongly connected.

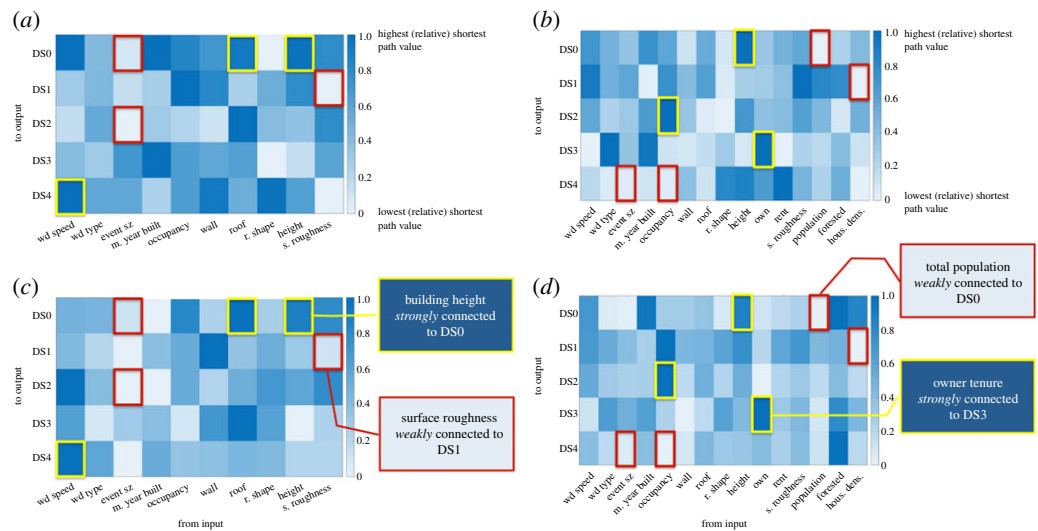

**Figure 5.** Shortest path relative values for (*a*) Model (A) combined ANNs, (*b*) Model (A) averaged results from each ANN, (*c*) Model (B) combined ANNs and (*d*) Model (B) averaged results from each ANN.

In the interest of accurate evaluation, strong (or weak) connections needed to be apparent in both the averaged and combined evaluation approaches. Figure 5 shows the respective relative results in calculating the shortest path for Model (A) and Model (B). By analysing both models, the effect of including social factors could be assessed as well. In Model (A), the roof material and building height showed a strong connection to DS0. This was considered reasonable due to the fact that the wind pressure increases with height and when wind damage starts it tends to be at the roof level. The wind speed was also found to be strongly tied to DS4, which was to be expected given that at a certain wind speed, the rest of the factors would become less vital in determining the resulting damage state. Additionally, weaker connections were found between the event size and DS0 and DS2, as well as the connection between surface roughness and DS1. Once tenure and density/debris factors were introduced to the network for Model (B), a shift occurred from strong connections between structural aspects to stronger connections related to *how* the structure was used. While a strong connection still remains between building height and DS0, building occupancy and tenure have now become strong contributors to DS2 and DS3, respectively. *This specific shift highlights a potential important characteristic: that the use of a structure becomes vital at the line between moderate and repairable damage (DS2) and damage so extreme that an insurance company would consider it a total write off (DS3).* Another connection shift is that housing density has taken the place of surface roughness in being weakly connected to DS1, and event size has shifted from being weakly connected to DS0 and DS2 to being weakly connected to DS4. These shifts could indicate that the event's size (areal extent) contributes more to lower damage states and housing density (or debris potential) takes its place at higher damage states, which could illustrate a criticality of density over size for worsening damage.

Following the shortest path analysis, the centrality analysis, through the concepts of closeness and degree, related how connected a neuron is within the network as a whole. What ultimately emerged from this analysis was the ability to compare the organizational network structure. Figure 6 shows the comparison of Models (A) and (B), respectively, for the centrality results. By plotting the closeness versus degree, a linear relationship, or apparent network organization, may form if a high degree corresponds to a high closeness. Interestingly, Model (A) appeared less organized than Model (B), which showed distinct alignment of input and output variables. What is most intriguing about this is that the knowledge within the field of natural hazards suggests Model (B) should perform better than Model (A), but these centrality scores highlight the actual structural difference between the two network models despite their similarly low build errors. Such low building errors may imply applicability of ANNs for this modelling problem, but this may in fact NOT be the case. Additionally, one may expect Model (A) to perform better, as well as show more organization, since this model contains all the variables considered in determining damage states post-disaster. However, as this has not been shown to be the case, perhaps the additional variables in Model (B) are being taken into account when conducting damage surveys, but just not explicitly or consciously, and therefore not making it into our current damage models.

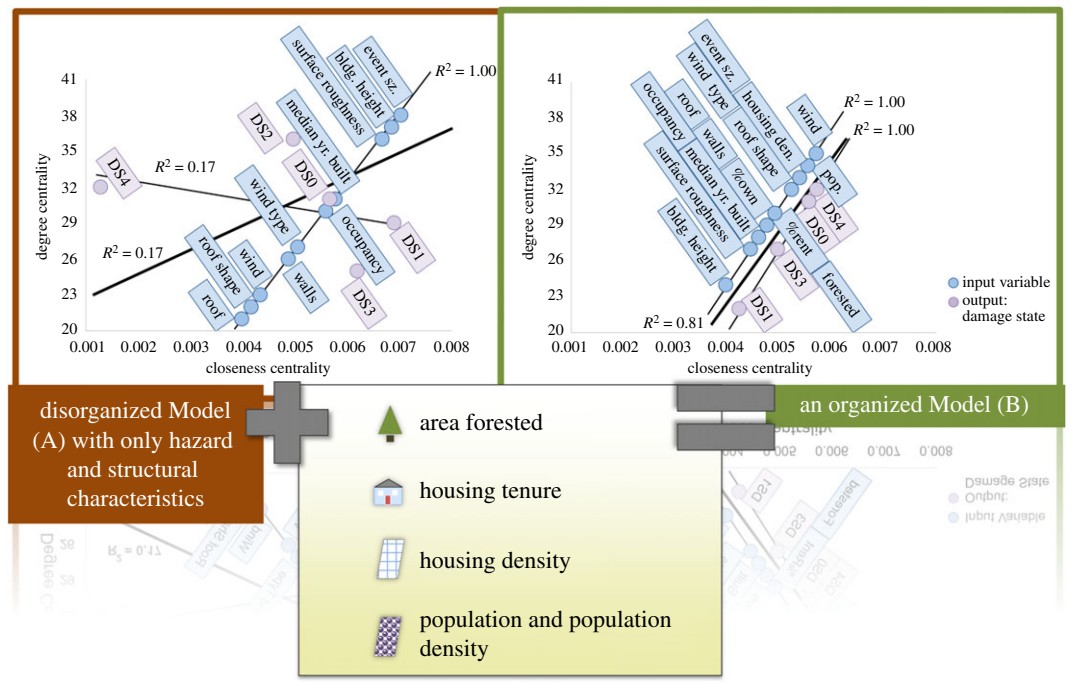

**Figure 6.** The effect of adding social and debris input variables into Model (A) to create Model (B) as it pertains to network centrality scores.

# 6. Conclusion

In using ANNs to model topics where the conceptual results are also understood, graph theory could provide means by which to interpret what is occurring within the 'black box' structure. The shortest path analysis shift from structural characteristics to building occupancy and use importance aligns with the concept of infrastructure maintenance and code compliance contributing to structural failures during extreme events. However, the shortest path analysis may be better served in determining exactly how multi-discipline variables contribute to specific outcomes, after the centrality score comparison shows organization within the network as a form of model validation. The significant shift in best-fit linear relationships between centrality scores brings to light a means to determine if an ANN is operating with conceptual accuracy instead of assuming an ANN model is acceptable for use due to its low build error. However, future research using similar methods would be recommended to further corroborate this argument. Further graphical analysis for this same modelling problem but instead using multiple hidden layers (or deep learning) could support such findings. Additionally, applying similar graph theory concepts to other machine learning models where the problem is also well understood could help support the use of such methods to ensure best practice measures in applying ANNs. Such graphical analysis may also provide means in better understanding how to intertwine topics that have proved conceptually difficult in the past. If further verified, the use of graph theory to understand ANNs could provide transparency through creating interpretable machine learning models as well as a means to better understand and uncover technical, topic-specific, concepts we may have been missing.

Data accessibility. See the electronic supplementary material for a full description of the data. Data and the associated codes are available at the Dryad Digital Repository at: https://doi.org/10.5061/dryad.f7m0cfxsk [58].

Authors' contributions. S.F.P. and H.N.M. conceived and designed the study. S.F.P. collected and analysed the data and wrote the initial draft of the paper. H.N.M. supervised the work and edited the manuscript.

Competing interests. We declare we have no competing interests.

Funding. This study was funded as part of cooperative agreement no. 70NANB15H044 between the National Institute of Standards and Technology (NIST) and Colorado State University and is gratefully acknowledged. The content expressed in this paper are the views of the authors and do not necessarily represent the opinions or views of CSU, NIST or the US Department of Commerce.

Acknowledgements. The authors wish to thank the National Institute of Standards and Technology for funding this research. We also wish to thank the Associate Editor, Professor Weisi Guo and the Subject Editor, Professor R. Kerry Rowe for handling the paper.

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
