## [Reviewer comments · Royal Society Open Science]

Review History

RSOS-200922.R0 (Original submission)

Review form: Reviewer 1

Is the manuscript scientifically sound in its present form?

Yes

Are the interpretations and conclusions justified by the results?

Yes

Is the language acceptable?

Yes

Do you have any ethical concerns with this paper?

No

Have you any concerns about statistical analyses in this paper?

No

Recommendation?

Accept with minor revision (please list in comments)

Comments to the Author(s)

The paper provides an interesting discussion on interpreting ANNs using graph theory, which seems to have been motivated by analogous research in the medical field interpreting MRIs using graph theory while explaining brain neurons and sensors. The paper is well-written and significant information is provided in the supplementary material. The reviewer has only few minor comments and suggestions to enhance the readability of the paper and flow of the discussions as follows:

- 1- It might be beneficial to elaborate in the introduction section on what "black box" attributes are being uncovered to provide the sought ANNs transparency and better explain the specific objectives of the study. For example, is mapping the shortest path an objective or rather shortest path is an encountered feature that is being explained in the context of relating ANNs inputs/outputs?
- 2- An additional paragraph at the end of section 1 that briefly outlines the content and different sections of the paper especially on what is provided in the paper body and what is in the supplementary materials can help the reader follow better.
- 3- In terms of reliance on supplementary material, there are few things that need to be still mentioned to some extent in the paper itself. Such as what platforms have been used for training and generating the ANNs general data format (e.g. what is the nature and format of each of the 117 points used in the training and testing), etc.)
- 4- In Page 9, line 55, the text is referring to 10 different algorithms among which the Bayesian has been selected. Only numbered references [46-55] of the 10 algorithms are listed, so I suggest that you briefly mention and discuss the 10 algorithms and specify why the Bayesian Regulation has been selected as this is an important point that can be ultimately tied to ANNs performance.
- 5- The reference to many figures in the text seem to be incorrect (typos most likely). For example, Page 7 line 29, the text is referring to Figure 5 but seems Figure 2 is what is meant. Same thing in page 14 (line 27) and page 15 (line 30) where the text is pointing to incorrect figures.

Decision letter (RSOS-200922.R0)

Dear Dr Mahmoud

On behalf of the Editors, we are pleased to inform you that your Manuscript RSOS-200922 "Interpreting the Socio-Technical Interactions within a Wind Damage-Artificial Neural Network Model for Community Resilience" has been accepted for publication in Royal Society Open Science subject to minor revision in accordance with the referees' reports. Please find the referees' comments along with any feedback from the Editors below my signature.

Please submit your revised manuscript and required files (see below) no later than 7 days from today's (ie 02-Oct-2020) date. Note: the ScholarOne system will 'lock' if submission of the revision

is attempted 7 or more days after the deadline. If you do not think you will be able to meet this deadline please contact the editorial office immediately.

on behalf of Professor Weisi Guo (Associate Editor) and R. Kerry Rowe (Subject Editor)
openscience@royalsociety.org

Reviewer comments to Author:

Reviewer: 1
Comments to the Author(s)

The paper provides an interesting discussion on interpreting ANNs using graph theory, which seems to have been motivated by analogous research in the medical field interpreting MRIs using graph theory while explaining brain neurons and sensors. The paper is well-written and significant information is provided in the supplementary material. The reviewer has only few minor comments and suggestions to enhance the readability of the paper and flow of the discussions as follows:

1- It might be beneficial to elaborate in the introduction section on what “black box” attributes are being uncovered to provide the sought ANNs transparency and better explain the specific objectives of the study. For example, is mapping the shortest path an objective or rather shortest path is an encountered feature that is being explained in the context of relating ANNs inputs/outputs?

2- An additional paragraph at the end of section 1 that briefly outlines the content and different sections of the paper especially on what is provided in the paper body and what is in the supplementary materials can help the reader follow better.

3- In terms of reliance on supplementary material, there are few things that need to be still mentioned to some extent in the paper itself. Such as what platforms have been used for training and generating the ANNs general data format (e.g. what is the nature and format of each of the 117 points used in the training and testing), etc.)

4- In Page 9, line 55, the text is referring to 10 different algorithms among which the Bayesian has been selected. Only numbered references [46-55] of the 10 algorithms are listed, so I suggest that you briefly mention and discuss the 10 algorithms and specify why the Bayesian Regulation has been selected as this is an important point that can be ultimately tied to ANNs performance.

5- The reference to many figures in the text seem to be incorrect (typos most likely). For example, Page 7 line 29, the text is referring to Figure 5 but seems Figure 2 is what is meant. Same thing in page 14 (line 27) and page 15 (line 30) where the text is pointing to incorrect figures.

===PREPARING YOUR MANUSCRIPT===

- one version identifying all the changes that have been made (for instance, in coloured highlight, in bold text, or tracked changes);
- a 'clean' version of the new manuscript that incorporates the changes made, but does not highlight them. This version will be used for typesetting.

===PREPARING YOUR REVISION IN SCHOLARONE===

- An individual file of each figure (EPS or print-quality PDF preferred [either format should be produced directly from original creation package], or original software format).
 - An editable file of each table (.doc, .docx, .xls, .xlsx, or .csv).
 - An editable file of all figure and table captions.
- Note: you may upload the figure, table, and caption files in a single Zip folder.
- Any electronic supplementary material (ESM).
 - If you are requesting a discretionary waiver for the article processing charge, the waiver form must be included at this step.
 - If you are providing image files for potential cover images, please upload these at this step, and inform the editorial office you have done so. You must hold the copyright to any image provided.
 - A copy of your point-by-point response to referees and Editors. This will expedite the preparation of your proof.

- Ensure that your data access statement meets the requirements at <https://royalsociety.org/journals/authors/author-guidelines/#data>. You should ensure that you cite the dataset in your reference list. If you have deposited data etc in the Dryad repository, please only include the 'For publication' link at this stage. You should remove the 'For review' link.
- If you are requesting an article processing charge waiver, you must select the relevant waiver option (if requesting a discretionary waiver, the form should have been uploaded at Step 3 'File upload' above).
- If you have uploaded ESM files, please ensure you follow the guidance at <https://royalsociety.org/journals/authors/author-guidelines/#supplementary-material> to include a suitable title and informative caption. An example of appropriate titling and captioning may be found at https://figshare.com/articles/Table_S2_from_Is_there_a_trade-off_between_peak_performance_and_performance_breadth_across_temperatures_for_aerobic_scope_in_teleost_fishes_/3843624.

Author's Response to Decision Letter for (RSOS-200922.R0)

See Appendix A.

Decision letter (RSOS-200922.R1)

Dear Dr Mahmoud,

It is a pleasure to accept your manuscript entitled "Interpreting the Socio-Technical Interactions within a Wind Damage-Artificial Neural Network Model for Community Resilience" in its current form for publication in Royal Society Open Science.

on behalf of Professor Weisi Guo (Associate Editor) and R. Kerry Rowe (Subject Editor)
openscience@royalsociety.org

Appendix A

Royal Society Open Science Manuscript- Paper #RSOS-200922

Interpreting the Socio-Technical Interactions within a Wind Damage-Artificial Neural Network Model for Community Resilience

RESPONSE TO REVIEWER 1:

General Comments:

The paper provides an interesting discussion on interpreting ANNs using graph theory, which seems to have been motivated by analogous research in the medical field interpreting MRIs using graph theory while explaining brain neurons and sensors. The paper is well-written and significant information is provided in the supplementary material. The reviewer has only few minor comments and suggestions to enhance the readability of the paper and flow of the discussions as follows:

Authors Response: We sincerely appreciate all the time the reviewer spent in reading the manuscript and providing extensive and thoughtful comments. Addressing them in the revised manuscript, as noted below, has enabled us to improve its quality significantly.

We addressed all comments and provided detailed response to all the points raised. **Changes made to address the comments are underlined below and appear in the manuscript in blue font.**

Specific Comments:

- 1- It might be beneficial to elaborate in the introduction section on what “black box” attributes are being uncovered to provide the sought ANNs transparency and better explain the specific objectives of the study. For example, is mapping the shortest path an objective or rather shortest path is an encountered feature that is being explained in the context of relating ANNs inputs/outputs?

Authors Response: Thank you for raising these issues. We agree with the reviewer's point that adding this perspective and clarification. The objective of the study was to try to explain the physical connections between inputs and outputs in the ANNs and the shortest path is one way to do so. The manuscript has been modified to addresses the aspects raised above as follows:

[please see page 2 and 3]:

“Interpreting both of these ANNs using common mathematical treatment of graph theory (e.g. shortest path) may allow for potential insight into what connections are occurring to relate the various input parameters (e.g. wind hazard, structural parameters, and social characteristics) to outputs (structural damage) within the ANN “black box” and provide insight into how social and engineering characteristics are affecting an overall resulting outcome.

- 2- An additional paragraph at the end of section 1 that briefly outlines the content and different sections of the paper especially on what is provided in the paper body and what is in the supplementary materials can help the reader follow better.

Authors Response: Thank you. The manuscript has been modified and a paragraph is added at the end of the introduction section to explain the content and the flow of the paper.

[please see page 3]:

“In this paper, we first, in Section 2, provide a high-level overview of ANNs and highlight one of the major issues with their use in different disciplines, which is the ability to physically interpret the relationship between inputs and outputs in well-trained networks. In Section 3, we provide background on graph theory, its overall use in different engineering and medical fields, and its possible use for unraveling the connections between input variables and outcomes within an ANN. In Section 4, we focus on the research conducted with emphasis on data assimilation, building the ANNs, and selecting the ANN models for analysis. We finally provide discussion of the results in Section 5 followed by general conclusions in Section 6.”

- 3- In terms of reliance on supplementary material, there are few things that need to be still mentioned to some extent in the paper itself. Such as what platforms have been used for training and generating the ANNs general data format (e.g. what is the nature and format of each of the 117 points used in the training and testing), etc.)

Authors Response: Thank you for the comment. We have revised the manuscript in Section 4(i) on Assimilation of Data to provide more description in the data. The original manuscript had detailed description of the data with more information provided in the Supplementary Information. We revised the paper to clarify this information as noted below.

[please see page 6&7]:

1. Methods

(i.) Assimilation of Data

“To create the models discussed in this article, data assimilation was necessary from two very different subject areas (engineering and social vulnerability) at the time of a wind event. Indeed, the validity of the approach is reliant on what data is available. The National Weather Service (NWS) maintains publicly available records from damage surveys related to wind events [28]. These records typically comprise of building damage description, location, wind speed, date of event, and, often, a photo of the damaged structure. From this, the building construction (materials and geometries), hazard parameters, and damage state of the building were gathered and organized in a text file. The U.S. Census Bureau tracks many population characteristics, including factors related to social vulnerability [33–35]. This data can be spatially organized by (in decreasing area size) county, census tract, block group, and block.

In this study, 117 data points were collected. Each building data point was associated with structural and social characteristics as illustrated by example, for select features, in Figure 2. Building structural characteristics include median year built for buildings in the specified block group (from U.S. Census), occupancy class, main wind force resisting system (MWFRS) material and the exterior façade materials (both wall and roofing), the roof shape, number of (equivalent) stories, and the footprint area. These data variables were all chosen based on generally accepted factors used in calculating wind resistance of a structure [12, 26, 36, 37] and are mostly gathered from building images geotagged by an NWS damage survey. These buildings then assume the census block group demographic characteristics for its location. Of all available census data, the demographics were chosen based on the vulnerability factors outlined in various sociological studies [33–35, 38–41]. For this study, a preliminary data set was created for the state of Missouri as a baseline for determining what factors intertwine best to determine building damage. Missouri was largely chosen for its location within Tornado Alley and so that the models built could be validated against known data from the 2011 Joplin tornado. The full compiled Missouri data, the key for how these items were coded, and an overview of the comparable performances of multiple model variations are available in Section S1 of the Supplementary Information.”

- 4- In Page 9, line 55, the text is referring to 10 different algorithms among which the Bayesian has been selected. Only numbered references [46-55] of the 10 algorithms are listed, so I suggest that you briefly mention and discuss the 10 algorithms and specify why the Bayesian Regulation has been selected as this is an important point that can be ultimately tied to ANNs performance.

Authors Response: Thank you for your comment. All algorithms were explained in great details in Section S2 of the Supplementary Information including the error analysis which provides justifications for using the Bayesian model. We have revised the main paper to make it clear that such information is provided in the Supplementary Information as noted below.

[please see page 9]:

“Subsequently, the target value, Y , was a function of the data set $D(x,y)$ and the activation function, s . The error determined in Equation (8) was propagated back through the network to adjust the weights and biases as well as change the neuron activations by means of a myriad of training algorithm approaches. An analysis of approximately 10 algorithms [46–55] was conducted and Bayesian Regulation was determined to produce the lowest percent error during the build process, the details of which can be found in Section S2 in the Supplementary Information. This training algorithm is probabilistic focused in Bayes Theory, instead of using gradient descent type of approaches. In other words, BR evaluates the probability of the weights, w_{ij} , on the connections between neurons given the data set, D [54, 55] such that:”

“The results of these statistical distributions are shown in detail in Section S3 in the Supplementary Information. Ultimately, this analysis of the range of performance for different training algorithms and subsequently different model inputs resulted in an initial choosing of Model (A) as a better performing model and Model (B) as one of the poorer performing models. However, in the interest of further confirmation, both Models (A) and (B) were simulated for the 2011 Joplin tornado an compared to the known results.”

- 5- The reference to many figures in the text seem to be incorrect (typos most likely). For example, Page 7 line 29, the text is referring to Figure 5 but seems Figure 2 is what is meant. Same thing in page 14 (line 27) and page 15 (line 30) where the text is pointing to incorrect figures.

Authors Response: Thank you for your comment. The paper has been thoroughly checked and all Figures re now properly referenced.